# Variability in Breast Cancer Biomarker Assessment and the Effect on Oncological Treatment Decisions: A Nationwide 5-Year Population-Based Study

**DOI:** 10.3390/cancers13051166

**Published:** 2021-03-09

**Authors:** Balazs Acs, Irma Fredriksson, Caroline Rönnlund, Catharina Hagerling, Anna Ehinger, Anikó Kovács, Rasmus Røge, Jonas Bergh, Johan Hartman

**Affiliations:** 1Department of Oncology and Pathology, Karolinska Institutet, 17176 Stockholm, Sweden; balazs.acs@ki.se (B.A.); caroline.ronnlund@ki.se (C.R.); jonas.bergh@ki.se (J.B.); 2Department of Clinical Pathology and Cytology, Karolinska University Laboratory, 11883 Stockholm, Sweden; 3Department of Breast, Endocrine Tumors and Sarcoma, Karolinska University Hospital, 17176 Stockholm, Sweden; irma.Fredriksson@ki.se; 4Department of Molecular Medicine and Surgery, Karolinska Institutet, 17176 Stockholm, Sweden; 5Division of Clinical Genetics, Department of Laboratory Medicine, Lund University, 22185 Lund, Sweden; catharina.hagerling@med.lu.se (C.H.); anna.ehinger@med.lu.se (A.E.); 6Division of Oncology and Pathology, Department of Clinical Sciences Lund, Lund University, 22184 Lund, Sweden; 7Department of Clinical Pathology, Sahlgrenska University Hospital, 41345 Gothenburg, Sweden; aniko.kovacs@vgregion.se; 8Department of Clinical Medicine, Aalborg University, 9000 Aalborg, Denmark; rr@rn.dk; 9NordiQC, Institute of Pathology, Aalborg University Hospital, 9000 Aalborg, Denmark; 10Breast Center, Cancer Theme, Karolinska University Hospital and Karolinska Comprehensive Cancer Center, Gävlegatan 55, 17164 Solna, Sweden

**Keywords:** positivity rate, biomarker, breast cancer, endocrine treatment, HER2-targeted treatment, variability

## Abstract

**Simple Summary:**

Biomarkers that define breast cancer treatment recommendations include estrogen receptor (ER), progesterone receptor (PR), and human epidermal growth-factor receptor 2 (HER2); histological grade; and in many countries, the Ki67 proliferation index. However, the subjective nature and degree of variability in breast cancer biomarker assessment might result in under- or overtreatment. We demonstrated that limited variability exists in ER, PR, and HER2 positivity rates among 29 departments in Sweden, including 43,261 patients. However, even a few outlier labs affect endocrine and anti-HER2 treatment rates in a clinically relevant proportion, indicating a need for improvement. Despite international guidelines, standardized protocols, and external quality control procedures, very high variability was found in Ki67 scoring and histological grading, indicating a need for new methods. Monitoring rates of biomarker expression and treatments among departments should be mandatory in order to detect variability issues affecting the clinical management of breast cancer.

**Abstract:**

We compared estrogen receptor (ER), progesterone receptor (PR), human epidermal growth-factor receptor 2 (HER2), Ki67, and grade scores among the pathology departments in Sweden. We investigated how ER and HER2 positivity rates affect the distribution of endocrine and HER2-targeted treatments among oncology departments. All breast cancer patients diagnosed between 2013 and 2018 in Sweden were identified in the National Quality Register for Breast Cancer. Cases with data on ER, PR, HER2, Ki67, grade, and treatment were selected (43,261 cases from 29 departments following the guidelines for biomarker testing). The ER positivity rates ranged from 84.2% to 97.6% with 6/29 labs out of the overall confidence intervals (CIs), while PR rates varied between 64.8% and 86.6% with 7/29 labs out of the CIs. HER2 positivity rates ranged from 9.4% to 16.3%, with 3/29 labs out of the overall CIs. Median Ki67 varied between 15% and 30%, where 19/29 labs showed significant intra-laboratory variability. The proportion of grade-II cases varied between 42.9% and 57.1%, and 13/29 labs were outside of the CI. Adjusting for patient characteristics, the proportion of endocrine and anti-HER2 treatments followed the rate of ER and HER2 positivity, illustrating the clinical effect of inter- and intra-laboratory variability. There was limited variability among departments in ER, PR, and HER2 testing. However, even a few outlier pathology labs affected endocrine and HER2-targeted treatment rates in a clinically relevant proportion, suggesting the need for improvement. High variability was found in grading and Ki67 assessment, illustrating the need for the adoption of new technologies in practice.

## 1. Introduction

In the era of precision medicine, patient management of invasive breast cancer is largely based on high-quality tumor biomarker testing. Indeed, breast cancer care has been a pioneer in the use of molecular features of the tumor to guide clinical decision making. Beside stage and tumor size, the leading biomarkers that define breast cancer treatment recommendations are estrogen receptor (ER), progesterone receptor (PR), and human epidermal growth-factor receptor 2 (HER2); histological grade; and in many countries, the Ki67 proliferation index [1]. Furthermore, ER and HER2 status are strongly predictive for anti-estrogen therapies [2,3] and HER2-targeted therapies [4,5]. All these biomarkers are established by the pathological analysis of tumor tissue, which, according to international guidelines, is mandatory in the analysis of all diagnosed primary breast cancer cases [1]. In clinical practice, immunohistochemical (IHC) detection is the gold standard for assessing ER, PR, and Ki67 in routine breast cancer pathology [6,7]. HER2 is analyzed by IHC, and in equivocal cases by in situ hybridization (ISH) [8]. IHC and ISH methods have several advantages, including wide applicability and availability with a low cost, ensuring the assessment of only invasive tumor cells, which may be challenging—especially in small tumors [9]. However, several studies have demonstrated moderate inter-observer and inter-laboratory reproducibility in breast cancer biomarker assessment [10,11]. Therefore, tests with proven analytical and clinical validity as well as clinical utility [12] are critical, as false-negative tests result in withholding effective treatment, while false-positive tests lead to overtreatment with costly and ineffective therapy, simultaneously resulting in unwanted direct and long-term side effects [13,14]. Parameters that predominantly influence the IHC and ISH results include pre-analytical (type of biopsy, tissue handling), analytical (IHC and ISH protocol), interpretive, and scoring steps [15]. Standardized procedures should be used in all steps of biomarker testing. International and national guidelines for the analysis and interpretation steps have been in place for several years [16]. In addition, to ensure compliance with validated protocols for testing, it is recommended that laboratories participate in regular internal and external quality control procedures like UK-NEQAS-ICC [17] or NordiQC [18]. The quality control work may also include comparisons between local and central laboratories.

Although external quality control has an important role in maintaining analytical validity, it cannot be said to represent daily routine, where time-constrained limitation always applies, and several pathologists perform the diagnostic evaluation. In quality-control sets it is often the most experienced pathologist with the highest skills in the specific field who performs the assessment. Furthermore, it is very difficult to compare labs using different antibodies, detection systems, and protocols. For this reason, the monitoring of positivity rates has been proposed as a tool to identify laboratories with insufficient assays and a high yield of false-positive or false-negative results [19,20]. A comparison of department performances in breast cancer biomarker testing is necessary to ensure that daily diagnostic practice meets clinical requirements.

In this study we compared the 5-year distribution of ER, PR, and HER2; Ki67 proliferation index; and histological grade evaluation scores from daily clinical practice among pathology departments in Sweden. We also investigated how ER and HER2 positivity rates affect the use of ER- and HER2-targeted treatments among oncology departments.

## 2. Materials and Methods

### 2.1. Data Source and Study Population

We retrieved patient data from NKBC, the Swedish National Quality Register for Breast Cancer (https://statistik.incanet.se/brostcancer/), which contains detailed clinical data on patient and tumor characteristics, treatment, and follow-up for all newly diagnosed primary cases of in situ and invasive breast cancer diagnosed in Sweden since 2008 [21]. The registry database is considered 99% complete [21]. The reported pathology data contains biomarker assessment data from the surgical specimen and, if preoperative systemic treatment, from pre-treatment core needle biopsy [21].

For this study, all invasive breast cancer cases diagnosed between 2013 and 2018 in Sweden were identified, and cases with available data on ER, PR, HER2, Ki67, and histological grade were selected (*n* = 43,959). Only pathology departments with more than 450 diagnosed breast cancer cases in the given period of time were included in the study, resulting in 43,261 valid cases from 29 investigated departments across Sweden. In cases with primary surgery, biomarker status was analyzed on the surgical specimen (*n* = 38,076), while in cases with neoadjuvant treatment, biomarker status was performed on the pre-treatment core biopsy (*n* = 5185). All studied pathology departments follow the national guidelines of biomarker evaluation [22], and participate in external quality control, passing breast cancer biomarker runs (Appendix A). The distribution of breast cancer phenotypes across the regions of Sweden is considered homogenous [21]. For each included case we extracted registry data on patient characteristics (age, date of diagnosis, and date of surgery) and tumor characteristics (type of tissue specimen, tumor size, histologic grade, lymph node status, ER, PR, and HER2 status) as well as detailed treatment data. Treatment data on neoadjuvant and adjuvant endocrine and anti-HER2 therapies were handled as binary variables (yes or no).

### 2.2. Biomarker Assessment

According to the Swedish guidelines [22], ER and PR status are considered positive when ≥10% of tumor cells show ER- and PR-specific staining in tumor nuclei detected by IHC. Both ER and PR status were taken into account as binary variables, either positive (≥10%) or negative (<10%). HER2 status was considered either positive or negative, detected by IHC and/or ISH as recommended by the Swedish guidelines; IHC was performed first, followed by amplification testing in case of a 2+ IHC score. Between 2013–2018, the Swedish guidelines were following the 2013 ASCO/CAP guidelines in reporting HER2 status.

The Ki67 proliferation index was analyzed as a continuous variable and defined as the percentage of tumor cells with positive nuclear staining counted in a hot-spot with a minimum of 200 tumor cells (in accordance with Swedish guidelines [22]). Histological grade was performed on hematoxylin and eosin (HE)-stained slides according to the Nottingham grade scoring system [23]. As intra-tumoral heterogeneity critically affects histological grade scoring on core biopsy specimens, these were excluded from the study.

### 2.3. Quality Assurance of Pathology Data Reported to the NKBC

In order to ensure the validity of NKBC data, 8 pathology departments from 3 different regions were selected to compare NKBC data with original pathology reports. Two included breast cancer cases per month at each site were randomly selected for validation (*n* = 1076). During part of the studied period, Ki67 was reported as either high or low (two categories), while more recently this was changed to high, intermediate, or low (three categories). Therefore, we decided to take into account the reported percentage values only (Ki67 proliferation index). Validity in biomarker reporting was 95–98.5% and considered acceptable (relative percentage error rate: 1.5–5%, Appendix A).

### 2.4. Questionnaire on Analytical Procedures

Since the labs use different antibodies, detection systems, protocols, etc., information on the analytical factors that might affect the inter-laboratory variation of the biomarkers was investigated with a questionnaire sent out covering antibody clone, IHC platform, and protocol used during the studied period. We could not detect any specific analytical factors causing inter- and intra-laboratory variability (Appendix A).

### 2.5. Statistical Analysis

Separate analysis was performed for ER, PR, HER2, Ki67, histological grade status, endocrine treatment, HER2-tageted treatment, age, tumor size, and lymph node status as outcome measures. Kruskal–Wallis test with Dunn’s post-hoc analysis and χ^2^ tests were used to compare the departments with one another, resulting in a total of 406 comparisons per marker. To investigate interannual variability for each department (intra-laboratory variability), χ^2^ test was used. We applied Bonferroni correction in multiple comparisons. To interpret the results, error bars and bar charts with 95% CI levels were used, where the biomarkers’ rates per department were plotted with the overall national proportion represented with 95% confidence limits as target. In all statistical analyses, the level of significance was set at *p* < 0.05. For statistical analysis, SPSS 25 software was used (IBM, Armonk, NY, USA).

## 3. Results

### 3.1. Patient and Tumor Characteristics

In total, 43,261 patients were included, with biomarker assessment performed at 29 different laboratories. Patient and tumor characteristics for all included cases are listed in Table 1. The mean age was 63.5 years and the median tumor size was 15 mm. Three out of four patients were node-negative (72%) based on sentinel-node biopsy and/or axillary dissection due to the mammography screening introduced over 20 years. More than half of the tumors were categorized as grade II (51.4%), 87.9% as ER-positive, and 72.8% as PR-positive. HER2 was considered positive in 13.1%. The median Ki67 proliferation index was 22%. Endocrine treatment was given to 73.8% of the patients, and HER2-targeted therapy in 11.6%.

### 3.2. Variability of ER Positivity Rates and Endocrine Therapy

The distribution of ER-positive cases ranged from 84.2% to 97.6% among the departments. All 29 labs were compared to each other, encompassing 406 comparisons in total (inter-lab variability). We found that 88 out of 406 lab comparisons showed significantly different ER positivity rates, and 5 labs had a positivity rate outside of the overall CI (Figure 1A). Four labs overvalued, while one lab undervalued the overall CI of ER positivity rates. Considering intra-laboratory variation in ER scoring, 7 out of 29 labs showed interannual variation in ER positivity rates (Figure 2A, Appendix A).

We compared the rate of endocrine therapy among the oncology departments, and found that 179 out of 406 department comparisons were significantly different. Furthermore, 13/29 departments had a rate of endocrine therapy outside of the CI (Figure 1A). However, we also found statistically significantly different patient characteristics among departments with regard to age, tumor size, and lymph node status at diagnosis (Appendix A). Furthermore, there were regional differences in offering endocrine treatment for tumors 10 mm or smaller. Therefore, we applied a case selection including only patients aged 70 or less and with tumor sizes greater than 10 mm. Adjusting for differences in age and tumor size among departments, we found that the rate of endocrine therapy strictly followed that of ER positivity (6 out of 406 department comparisons were statistically significant) (Figure 1B).

### 3.3. Variability of PR Positivity Rates

Positivity rates for PR among the departments varied between 64.8% and 86.6%. Overall, 119 out of 406 lab comparisons were significant, and 7 labs showed positivity rates outside of the CI (Appendix A). Regarding intra-laboratory variability, 10 out of 29 labs showed significantly different PR positivity rates between 2013 and 2018 (Appendix A).

### 3.4. Variability of HER2 Positivity Rates and Anti-HER2 Treatment

The distribution of HER2-positive cases among departments ranged from 9.4% to 16.3%. Regarding inter-lab variability, 14 out of 406 lab comparisons were significantly different, and two labs overvalued, while one lab undervalued the overall CI of HER2 positivity rates (Figure 3A). Considering intra-laboratory variation in HER2 scoring, 10 out of 29 labs showed significant interannual variation (Figure 2B, Appendix A).

A comparison of the rate of HER2-targeted treatments among the oncology departments showed that 19 out of 406 department comparisons were significantly different. Furthermore, five departments had treatment rates outside of the CI (Figure 3A). Adjusting for differences in age and tumor size among departments, we found that HER2-targeted therapy strictly followed that of HER2 positivity, with only one oncology department outside of the overall CI and 3 out of 406 comparisons statistically significantly different (Figure 3B).

### 3.5. Variability of Ki67 Proliferation Index

The median Ki67 proliferation index varied between 15% and 30% among the departments. Overall, 257 out of 406 lab comparisons were significant (Figure 4A). Regarding intra-laboratory variability, 19 out of 29 labs showed significant interannual variation (Figure 4B).

### 3.6. Variability of Histological Grade

The proportion of histological grade II cases among the departments ranged from 42.9% to 57.1%. Of 406 lab comparisons, 104 were significantly different, and 13 labs had a distribution rate outside of the overall CI (Figure 5A). When considering histological grade as a binarized variable, similar inter-lab variability was seen (93 out of 406 lab comparisons with *p* < 0.05). Considering intra-laboratory variation, 15 out of 29 labs showed significant interannual variation (Figure 5B).

## 4. Discussion

We investigated inter- and intra-laboratory variability in the assessment of ER, PR, HER2, Ki67, and histological grade in a non-selective population-based nationwide cohort of 43,261 invasive breast cancer patients diagnosed between 2013 and 2018, using real-world data from the Swedish National Quality Register for Breast Cancer (NKBC). The patients involved in this study represent 98.4% of breast cancer cases occurring between 2013 and 2018 in Sweden. We demonstrated that there was limited variability in ER, PR, and HER2 positivity rates among pathology departments. Overall positivity rates were 87.9% for ER, 72.8% for PR, and 13.1% for HER2, with a median Ki67 proliferation index of 22% and 51.4% of tumors deemed grade II, which are in line with international findings [19,20,24,25,26,27,28,29,30]. Although the proportion HER2 positivity ranged from 9.4% to 16.3% (thus varying from 1 positive out of 6 tested patients to 1 positive out of 11 patients tested), the confidence intervals were broadly overlapping for the vast majority of the labs. The same observation applies for positivity rates of ER and PR among the pathology departments. A recent investigation involving 33,046 patients performed in the Netherlands using a similar study design showed comparable variability among pathology departments in ER, PR, and HER2 positivity rates [25]. Other studies have shown significant variation among pathology laboratories, with HER2 positivity rates varying from 7.6% to 31.6% [19,20] and with significant outliers even after adjusting for population characteristics [20]. Moreover, our results demonstrate that when adjusting for clinicopathological factors, the distribution of endocrine therapy and HER2-targeted treatments strictly followed the differences of ER and HER2 positivity rates among pathology departments, leading to the same relative difference in drug prescription rates among oncology departments.

To the best of our knowledge, this is the first study to investigate the association between positivity rates of ER and HER2 among pathology laboratories and the distribution of endocrine therapy and HER2-targeted treatments among oncology departments.

Although the number of outlying departments is limited (ER: 6/29; PR: 7/29; HER2: 3/29), the slightly higher rate of intra-laboratory variability shows that there is still room for improvement—especially because such variation will be directly translated into treatment decisions. Stage, grade, and biomarker assessment are pillars for the multidisciplinary recommendations regarding surgery, chemotherapy, endocrine, and targeted therapies. Valid analyses with low variability could potentially improve patient outcomes.

Two previous studies investigated reproducibility in routine breast cancer bio-marker assessment at Swedish pathology departments. In the first study, a tissue microarray of 11 breast cancers to stain and evaluate for HER2 status was sent out twice to 24 departments [31]. The authors demonstrated a very good reproducibility in this round robin study, and showed a good analytical variability. However, the study did not reflect daily diagnostic routine [31]. In the second study, 10 cases from each of 27 participating departments were systematically collected and sent to a reference lab for re-staining and re-scoring of ER, PR, HER2, and Ki67 [32]. Although the authors demonstrated a very good agreement and observed discrepancies seemed to be explained by analytical differences, the study design did not enable comparison of individual departments [32]. In the present study, limited variability regarding ER, PR, and HER testing was found, which is in accordance with previous studies. On the other hand, we also demonstrated unacceptable inter- and intra-laboratory variability in histological grade and Ki67 assessment, urging imminent efforts to further standardize testing procedures. In a large study of 33,043 non-selected breast cancer patients, van Dooijeweert et al. also demonstrated a substantial variability in histological grading among pathology departments [29]. It has long been acknowledged that both Ki67 and histological grade provide useful prognostic information [30,33,34]. However, they are subject to reproducibility issues which diminish clinical utility [11,35,36]. According to the Swedish guidelines, all ER-positive breast cancer patients are categorized into luminal-A-like and luminal-B-like subtypes based on tumor histological grade, Ki67, and PR status. All grade I and grade III cases are categorized as luminal-A-like and luminal-B-like, respectively. Grade II cases with low and high Ki67 values (based on local cut-offs) are divided into luminal-A-like and luminal-B-like, respectively. Grade II cases with intermediate Ki67 score are further measured by PR status in order to being divided into luminal-A-like (PR expression ≥20%) and luminal-B-like (PR expression <20%) subtypes [37]. As both histological grade and Ki67 status play important roles in the prognostication of breast cancer patients, the high variability found between pathology departments regarding these markers has serious clinical implications. Although significant efforts have been made to standardize pathologist-read scoring for these markers [38], introducing new methods to this field such as digital-image analysis and molecular methods would hold the promise of increasing reproducibility [39,40,41,42,43]. Besides, we believe that Ki67 IHC and its evaluation should undergo rigorous external quality-control procedures (similarly to ER, PR, and HER2), which is not currently the case. Furthermore, the new International Ki67 in Breast Cancer Working Group recommendations have been established, providing specific and detailed guidelines for each step of Ki67 assessment with the levels of evidence for its technical validity and clinical utility as a biomarker [44]. Following these guidelines in clinical practice has the potential to reach an acceptable reproducibility for Ki67 assessment.

There are several potential limitations to this study. First of all, as the proportion of neoadjuvant treatment differs among oncology departments, the proportion of cases analyzed based on pretreatment biopsy instead of surgical specimen differ. Tumor heterogeneity might affect the concordance in biomarker expression rates between core biopsy sample and resection specimen. However, several studies have demonstrated good agreement in biomarkers between core biopsy sample and resection specimen, especially for ER [45,46,47]. However, this does not apply to histological grade, as only a small tumor area is presented in the biopsy specimen. For this reason, we excluded all grade scores reported based on core biopsy specimens only. Additionally, although the distribution of breast cancer molecular phenotypes seems to be homogenous between regions in Sweden, some differences in case-mix may remain despite adjustments for age and stage among different departments. Furthermore, the study design did not enable us to control the pre-analytical and analytical factors, potentially affecting assessments. For this reason, a questionnaire regarding used antibodies, IHC platforms, and protocols was sent out, and results from the NordiQC were retrieved to make sure that all participating departments had previously passed analytical validity testing. As IHC and ISH staining needs to be run overnight in practice, we hypothesized that the definitive biomarkers status might have been added to the pathology report afterward, and thus be missing in the report to NKBC. However, we found low rates of missing data, and it equally affected receptor-positive and receptor-negative tumors.

## 5. Conclusions

We demonstrated in a nation-wide population-based cohort study that there is limited variability in ER, PR, and HER2 testing among laboratories. However, as even a few outlier pathology labs affected endocrine and anti-HER2 treatment rates in a clinically relevant proportion, our study results indicate a need for further improvement. Despite international and national guidelines for analysis and interpretation, standardized protocols, and previous internal and external quality control procedures, very high variability was found in Ki67 scoring and histological grading, which suggests a need for the use of new methods. Monitoring and comparing rates of biomarker expression and treatments among departments should be mandatory in order to detect variability issues affecting the clinical management of breast cancer.

## Figures and Tables

**Figure 1 cancers-13-01166-f001:**
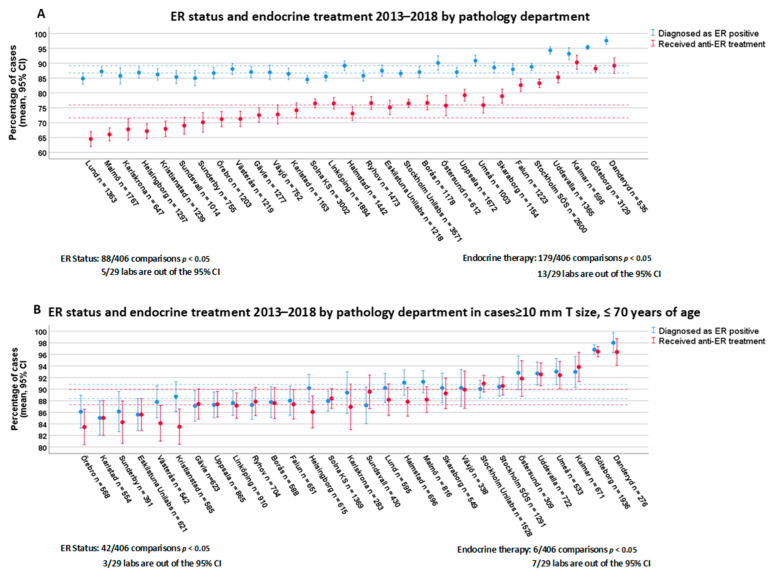
Endocrine receptor (ER) status and endocrine treatment between 2013 and 2018 by department. Dashed lines represent the CI for overall positivity (blue) and overall treatment rate (red). (**A**) includes all the patients while (**B**) shows results with cases ≥10 mm T size, ≤70 years of age.

**Figure 2 cancers-13-01166-f002:**
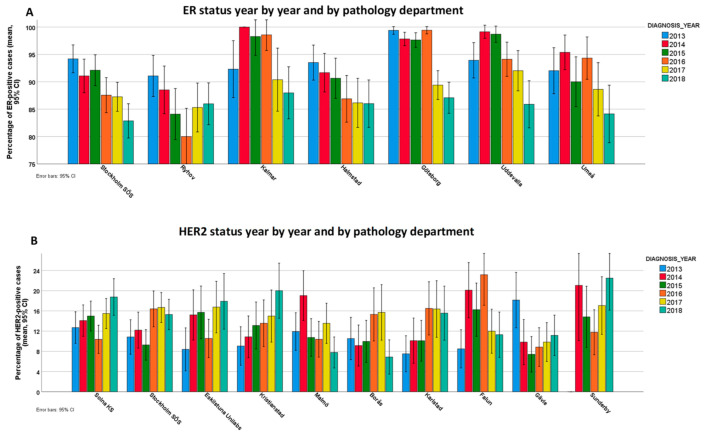
Intra-lab variability for ER (**A**) and human epidermal growth-factor receptor 2 (HER2) (**B**) status among pathology departments. Only laboratories showing statistically significant differences are shown. Please refer to Appendix A to see all the labs.

**Figure 3 cancers-13-01166-f003:**
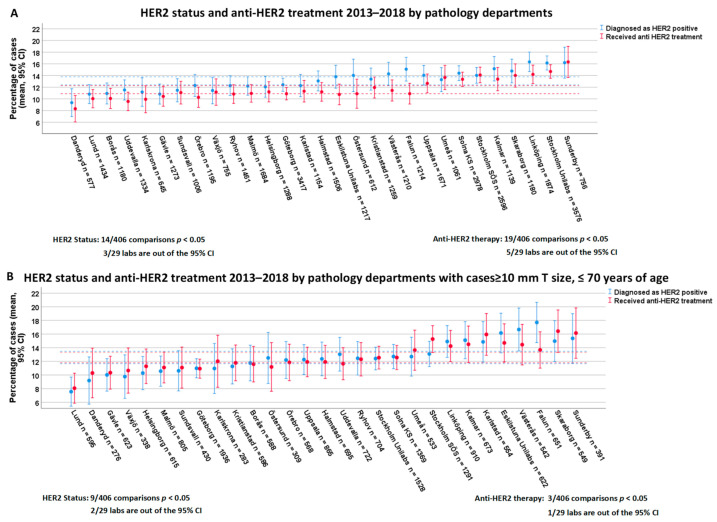
HER2 status and anti-HER2 treatment between 2013 and 2018 by pathology department. Dashed lines represent the CI for overall positivity (blue) and overall treatment rate (red). (**A**) includes all the patients while (**B**) shows results with cases ≥10 mm T size, ≤70 years of age.

**Figure 4 cancers-13-01166-f004:**
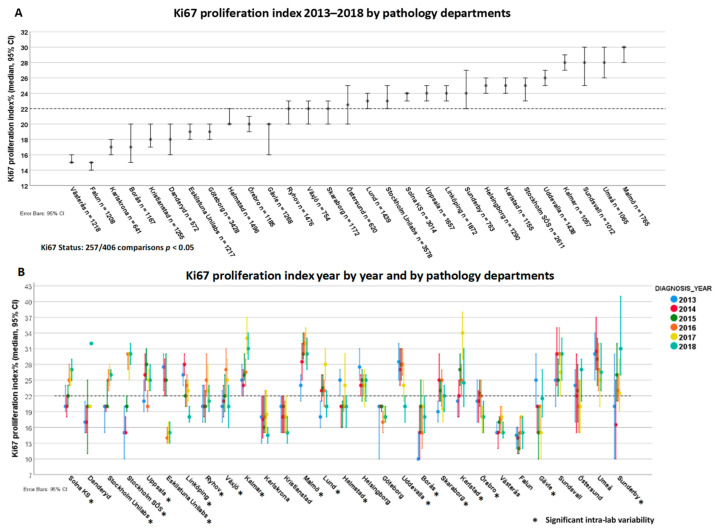
Inter- (**A**) and intra-laboratory variability (**B**) for Ki67 among 29 pathology departments.

**Figure 5 cancers-13-01166-f005:**
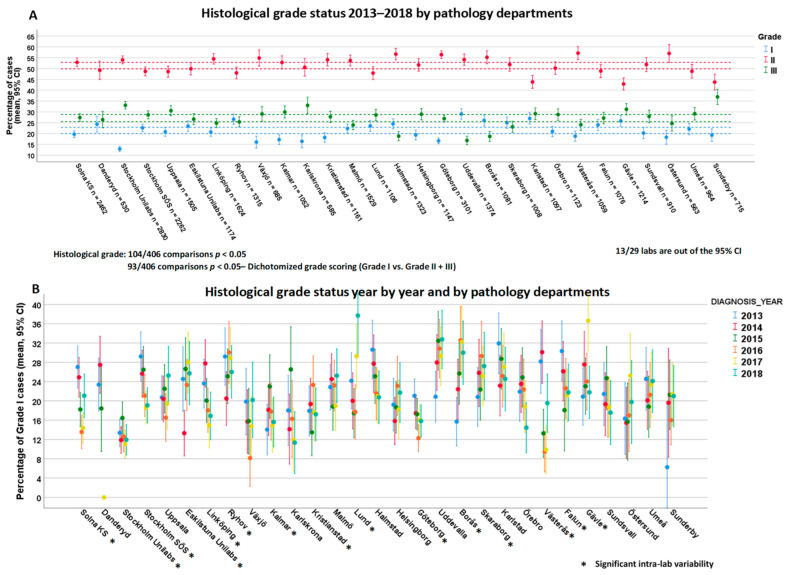
Inter- (**A**) and intra-laboratory variability (**B**) for histological grade status among 29 pathology departments.

**Table 1 cancers-13-01166-t001:** Patient and tumor characteristics and treatment in 43,261 cases with biomarker assessment performed at 29 pathology departments.

Variable	Category	Statistics	Results	
Patients		Total *n*, %	43,959	100
		Valid *n*, %	43,261	98.4
Age (years)		Mean ± SD	63.5	± 13.3
	Missing data	%	0.1	-
Tumor size (mm)		Median, IQR	15	13
	Missing data	%	1.1	-
Lymph node status	Node-negative	%, CI	72.0	71.0–73.0
	Node-positive	%, CI	28.0	27.0–29.0
	Missing data	%	2.3	-
Histological grade (*n* = 38,076)	Grade I	%, CI	21.4	20.0–22.9
	Grade II	%, CI	51.4	49.9–52.9
	Grade III	%, CI	27.2	25.5–28.8
	Missing data	%	1.3	-
ER	Positive	%, CI	87.9	86.7–89.2
	Negative	%, CI	12.1	10.8–13.2
	Missing data	%	4.4	
PR	Positive	%, CI	72.8	70.9–74.8
	Negative	%, CI	27.2	25.2–29.1
	Missing data	%	5.7	-
HER2	Positive	%, CI	13.1	12.4–13.8
	Negative	%, CI	86.9	86.2–87.6
	Missing data	%	2.3	-
Ki67		Median, IQR	22.0	26
	Missing data	%	2.0	-
Endocrine treatment	Received	%, CI	73.8	71.6-75.9
	Not received	%, CI	26.2	24.1–28.4
	Missing data	%	0.7	-
HER2-targeted therapy	Received	%, CI	11.6	10.9–12.3
	Not received	%, CI	88.4	87.7–89.1
	Missing data	%	0.7	-

CI = 95 % confidence interval; IQR = interquartile range; SD = Standard deviation.

## Data Availability

Data are available on the website of Swedish National Quality Register for Breast Cancer (NKBC) and can be accessed via this link: https://statistik.incanet.se/brostcancer/.

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
