# Peer review of "Variability in Breast Cancer Biomarker Assessment and the Effect on Oncological Treatment Decisions: A Nationwide 5-Year Population-Based Study"

_cancers, 2021, doi:10.3390/cancers13051166_

Round 1

Reviewer 1 Report

An excellent paper. Accept as submitted.

Author Response

Thank you for your evaluation.

Reviewer 2 Report

The study aims to evaluate the variability in breast cancer diagnosis, across the pathology departments. This is an interesting study that needs to be performed periodically to evaluate the accuracy of methodologies and the need of reconsideration of clinical guidelines. 

Comments and suggestions
The statistical methods used, although simple, are correct and sufficient for the aim of this study. However, additional value can be added, and additional comparisons can be performed. The work would be improved if the association between survival and variability is tested, I mean, is there some association between the intralab variability and survival? It implies that those outlier labs have better or worst diagnosis accuracy, and perhaps this fact is affecting survival rates?
Another interesting analysis to perform, given the case that authors have access to this kind of data, is to evaluate the concordance level between IHC subtypes and molecular subtypes obtained with PAM50 or other available tests. I am not sure if this is possible, but if some labs have access to molecular subtypes or gene expression tests, this subset of samples can be used to evaluate the rate of concordance between these, and to evaluate an additional factor that may explain the variability.

The reasons of this intra-lab or inter-lab variability (specially in grade and ki67) should be further disscussed by authors (I find this very interesting!), and perhaps try to identify or propose weaknesses and strengths of the methodology, and whether a gradual change in this methodology (IHC) is feasible given the local circumstances or not.

Minor changes

Figure 2 is a litlle confusing and overwhelming, perhaps another type of graphic can bemore friendly to the reader, such as line graph, or to add more space between hospital bars. Same happens with Figure 5B, though this one is easier to reas.

Some of the material in Supplemental Material (page 15-15) can be summarized in a single table. 

There are a few typos that should be reviewed, but the overall writing is clear and of  good quality.

Author Response

Reviewer 2 Comments and Suggestions for Authors

The study aims to evaluate the variability in breast cancer diagnosis, across the pathology departments. This is an interesting study that needs to be performed periodically to evaluate the accuracy of methodologies and the need of reconsideration of clinical guidelines. 

Comments and suggestions
The statistical methods used, although simple, are correct and sufficient for the aim of this study. However, additional value can be added, and additional comparisons can be performed. The work would be improved if the association between survival and variability is tested, I mean, is there some association between the intralab variability and survival? It implies that those outlier labs have better or worst diagnosis accuracy, and perhaps this fact is affecting survival rates?

Thank you for this suggestion. We will definitely intend to perform this analysis in the future. However, in the current study, the length of follow-up time did not allow to investigate this with proper statistical confidence. The vast majority of the cases are luminal-like cancers with usually late recurrence (5+ years).

Another interesting analysis to perform, given the case that authors have access to this kind of data, is to evaluate the concordance level between IHC subtypes and molecular subtypes obtained with PAM50 or other available tests. I am not sure if this is possible, but if some labs have access to molecular subtypes or gene expression tests, this subset of samples can be used to evaluate the rate of concordance between these, and to evaluate an additional factor that may explain the variability.

Unfortunately, PAM50 (Prosigna) has been being used for less than a year in the clinical routine in Sweden and only for strictly selected cases (clinically an intermediate risk of recurrence). Therefore, we do not have PAM50 data for the current patient cohort. Thank you for this suggestion, we will update the investigation according to this in the near future.

The reasons of this intra-lab or inter-lab variability (specially in grade and ki67) should be further disscussed by authors (I find this very interesting!), and perhaps try to identify or propose weaknesses and strengths of the methodology, and whether a gradual change in this methodology (IHC) is feasible given the local circumstances or not.

We agree with the reviewer, please find a more detailed discussion on this matter on page 13, line 294-314. 

Minor changes

Figure 2 is a litlle confusing and overwhelming, perhaps another type of graphic can bemore friendly to the reader, such as line graph, or to add more space between hospital bars. Same happens with Figure 5B, though this one is easier to reas.

Some of the material in Supplemental Material (page 15-15) can be summarized in a single table. 

There are a few typos that should be reviewed, but the overall writing is clear and of  good quality.

We agree with the Reviewer, please see the updated figure 2 on page 8. The previous version has been put to the supplemental materials.

Reviewer 3 Report

Thank you for the opportunity to review this interesting manuscript which uses real-world data from Sweden to examine the issue of intra- and inter-laboratory variation in biomarker assessment in breast cancer.

The manuscript is well-written and clear. It has has the strengths of being a large dataset from the whole country. Appropriate validation has been carried out to confirm the accuracy of the centrally compiled data. The general consistency of ER/PR/HER2 reporting and the variation in tumour grade and Ki67 between laboratories is not surprising in light of known data but this is a large series of "real-world" cases.

The overall rate of ER positivity seems extremely high in this series - 87.9% (range 84.2-97.6%). Figure 2A suggests that in one centre (Danderyd) 100% of cases were ER positive in 2017 and 2018 - is this correct? It seems to me improbable than only 12.1% of breast cancer cases in Sweden from 2013-2018 were either ER- HER2+ or TNBC. Could the authors comment on this in the discussion?

Figure 2 is quite hard to read. I understand that the authors are presenting ER/HER2 status by year for each individual pathology department in the study but it's very difficult to interpret! Could the authors consider presenting this data in any other way - or at least separating out the departments with significant variability from those without? Also, most of the departments with significant variation in ER status appear to show a trend to a reducing proportion of ER+ cancers in later years - is this correct? If so, can the authors offer any explanation?

Unfortunately (and perhaps unsurprisingly) the cited guidelines for biomarker assessment are in Swedish (which is fair enough) and I can't read them. In the 2013 ASCO/CAP pathology reporting guidelines for HER2 status there was an ISH equivocal category (HER2/CEP17 ratio of 1.8-2.2). Could the authors clarify whether the Swedish guidelines have a similar category or whether this is a dichotomous report of amplified/non-amplified?

There is some variation in the use of endocrine therapy in ER+ disease which appears to be accounted for by ET not being used for T1a/b ER+ cancers. What are the Swedish guidelines on the use of ET in small cancers? Could the authors explain why there are regional differences in the use of ET for these tumours?

There is considerably variation in tumour grade, which is obviously a subjective tumour assessment. Do the authors have any information on the number of pathologists assessing breast specimens at each centre, and their level of experience? For example, did centres with fewer pathologists report less intra-laboratory variation?

The authors touch on the variation in the use of neoadjuvant therapy in the discussion section and reporting on core biopsy specimens may well account for some of the variation in grade. Can they correlate rate of neoadjuvant therapy use with intra-laboratory variation in grade to support this contention? This information is presumably available from the Swedish Quality Register used for the study.

Is Ki67 assessment routinely used for clinical decision making in Sweden? Could the authors touch on its use and the clinical implications of this variation in their discussion section?

Finally - do the authors have any suggestions as to how this variability can be addressed?!

Author Response

Reviewer 3 Comments and Suggestions for Authors

Thank you for the opportunity to review this interesting manuscript which uses real-world data from Sweden to examine the issue of intra- and inter-laboratory variation in biomarker assessment in breast cancer.

The manuscript is well-written and clear. It has has the strengths of being a large dataset from the whole country. Appropriate validation has been carried out to confirm the accuracy of the centrally compiled data. The general consistency of ER/PR/HER2 reporting and the variation in tumour grade and Ki67 between laboratories is not surprising in light of known data but this is a large series of "real-world" cases.

The overall rate of ER positivity seems extremely high in this series - 87.9% (range 84.2-97.6%). Figure 2A suggests that in one centre (Danderyd) 100% of cases were ER positive in 2017 and 2018 - is this correct? It seems to me improbable than only 12.1% of breast cancer cases in Sweden from 2013-2018 were either ER- HER2+ or TNBC. Could the authors comment on this in the discussion?

Yes, it is correct. The reason for 100% positivity rate for “Danderyd” is that breast pathology was moved from that site during the studied years, with very few included cases. The years with very few cases were not included in the analyses for this lab. This has now been highlighted in the figure legends of supplementary figure 1. Generally, the high number of ER positive cases, our finding is in line with other large series of clinical cases [1, 2]. The potential reason for this could be that the clinical trials investigated a very selected patient cohort, where these subtypes were enriched due to clinical importance.

Figure 2 is quite hard to read. I understand that the authors are presenting ER/HER2 status by year for each individual pathology department in the study but it's very difficult to interpret! Could the authors consider presenting this data in any other way - or at least separating out the departments with significant variability from those without? Also, most of the departments with significant variation in ER status appear to show a trend to a reducing proportion of ER+ cancers in later years - is this correct? If so, can the authors offer any explanation?

We agree with the Reviewer, please see the updated figure 2 on page 8. The previous version has been put to the supplemental materials. Unfortunately, we don’t have an obvious explanation for this trend. This is one of the reasons why we sent out the questionnaire about analytical variability and run a systematic quality assurance on our data 

Unfortunately (and perhaps unsurprisingly) the cited guidelines for biomarker assessment are in Swedish (which is fair enough) and I can't read them. In the 2013 ASCO/CAP pathology reporting guidelines for HER2 status there was an ISH equivocal category (HER2/CEP17 ratio of 1.8-2.2). Could the authors clarify whether the Swedish guidelines have a similar category or whether this is a dichotomous report of amplified/non-amplified?

For this period of time, the Swedish guidelines were following the 2013 ASCO/CAP guidelines in reporting HER2 status. This sentence has now been added to the manuscript on page 3, line 129.

There is some variation in the use of endocrine therapy in ER+ disease which appears to be accounted for by ET not being used for T1a/b ER+ cancers. What are the Swedish guidelines on the use of ET in small cancers? Could the authors explain why there are regional differences in the use of ET for these tumours?

According to the Swedish treatment guidelines, decision on postoperative endocrine treatment should be based on assessment of risk of recurrence, including estimation of the risk-benefit ratio. In general, all patients with ER-positive breast cancer should be offered endocrine therapy. We have indications of regional differences; patients the Stockholm region with 20-25% of the Swedish population are routinely offered this therapy option, even for T1a/b cancers being ER positive. The relative benefit of postoperative adjuvant endocrine therapy is equivalent regardless of lymph node status, age, progesterone receptor content and whether the patient has received chemotherapy or not. The absolute benefit, however, depends on the individual risk of recurrence why also factors like age and comorbidity can be taken into account. The guidelines therefore state that for patients with a very low risk of relapse (small luminal A-like breast cancers T1a-b), can one eventually refrain from endocrine adjuvant therapy after discussion with the patient about treatment benefits and risks. Some differences in regional interpretations of the national guidelines remain.

There is considerably variation in tumour grade, which is obviously a subjective tumour assessment. Do the authors have any information on the number of pathologists assessing breast specimens at each centre, and their level of experience? For example, did centres with fewer pathologists report less intra-laboratory variation?

We don’t know the exact number of pathologists at every site, but each case is always signed out by a specialist with special focus on breast cancer (breast pathologist). Usually this means that approximately 2-3 pathologists per site assess the cases.

The authors touch on the variation in the use of neoadjuvant therapy in the discussion section and reporting on core biopsy specimens may well account for some of the variation in grade. Can they correlate rate of neoadjuvant therapy use with intra-laboratory variation in grade to support this contention? This information is presumably available from the Swedish Quality Register used for the study.

Due to the relatively low number of patients received neoadjuvant therapy compared with adjuvant treatment per site, we could not correlate this context with grade. Furthermore, we decided to exclude grade scores reported only from core biopsies to decrease the potential variated between labs in grade scoring. This has now been added to the text on page 3, line 135 and page 14, line 322.

Is Ki67 assessment routinely used for clinical decision making in Sweden? Could the authors touch on its use and the clinical implications of this variation in their discussion section?

Yes, all ER positive breast cancer patients in Sweden are according to the Swedish guidelines categorized into Luminal A-like and Luminal B-like subtypes based on histological grade, Ki67 and PR status. All grade I and grade III cases are categorized as Luminal A-like and Luminal B-like, respectively. Grade II cases with low and high Ki67 values (based on local cut-offs) are divided into Luminal A-like and Luminal B-like, respectively. Grade II cases with intermediate Ki67 score are further measured by PR status in order to being divided into Luminal A-like (≥ 20% PR expression) and Luminal B-like (< 20% PR ex-pression) subtypes [3]. Due to this, assessment of Ki67 has significant clinical implications which has now been emphasized in the discussion on page 13, line 294-314.

Finally - do the authors have any suggestions as to how this variability can be addressed?!

We believe that introducing new methods to this field such as digital-image analysis and molecular methods would hold the promise to increase reproducibility for both Ki67 and histological grade. Besides, Ki67 IHC and its evaluation should undergo rigorous external quality control procedures which might also aid this problem. The new International Ki67 in Breast Cancer Working Group recommendations (published in 2020 December) provides the standards to potentially reach an acceptable variation in Ki67 scoring [4].